# SAR296968, a Novel Selective Na^+^/Ca^2+^ Exchanger Inhibitor, Improves Ca^2+^ Handling and Contractile Function in Human Atrial Cardiomyocytes

**DOI:** 10.3390/biomedicines10081932

**Published:** 2022-08-09

**Authors:** Philipp Hegner, Marzena Drzymalski, Alexander Biedermann, Bernadette Memmel, Melanie Durczok, Michael Wester, Bernhard Floerchinger, Zdenek Provaznik, Christof Schmid, York Zausig, Lars S. Maier, Stefan Wagner

**Affiliations:** 1Department of Internal Medicine II, University Medical Center Regensburg, 93053 Regensburg, Germany; 2Department of Cardiothoracic Surgery, University Medical Center Regensburg, 93053 Regensburg, Germany; 3Department of Anesthesiology, University Medical Center Regensburg, 93053 Regensburg, Germany; 4Department of Anesthesiology and Operative Intensive Care Medicine, Aschaffenburg-Alzenau Hospital, 63739 Aschaffenburg, Germany

**Keywords:** Na^+^/Ca^2+^ exchanger, NCX, HFpEF, SAR296968

## Abstract

Background: In reverse-mode, cardiac sodium-calcium exchanger (NCX) can increase the cytoplasmic Ca^2+^ concentration in response to high intracellular Na^+^ levels, which may contribute to diastolic contractile dysfunction. Furthermore, increased spontaneous Ca^2+^ release from intracellular stores can activate forward mode NCX. The resulting transient inward current causes delayed afterdepolarization (DAD)-dependent arrhythmias. Moreover, recently, NCX has been associated with impaired relaxation and reduced cardiac function in heart failure with preserved ejection fraction (HFpEF). Since NCX is upregulated in human chronic atrial fibrillation (AF) as well as heart failure (HF), specific inhibition may have therapeutic potential. Objective: We tested the antiarrhythmic, lusitropic and inotropic effects of a novel selective NCX-inhibitor (SAR296968) in human atrial myocardium. Methods and Results: Right atrial appendage biopsies of 46 patients undergoing elective cardiac surgery in a predominant HFpEF cohort (n = 24/46) were investigated. In isolated human atrial cardiomyocytes, SAR296968 reduced the frequency of spontaneous SR Ca^2+^ release events and increased caffeine transient amplitude. In accordance, in isolated atrial trabeculae, SAR296968 enhanced the developed tension after a 30 s pause of electrical stimulation consistent with reduced diastolic sarcoplasmic reticulum (SR) Ca^2+^ leak. Moreover, compared to vehicle, SAR296968 decreased steady-state diastolic tension (at 1 Hz) without impairing developed systolic tension. Importantly, SAR296968 did not affect the safety parameters, such as resting membrane potential or action potential duration as measured by patch clamp. Conclusion: The novel selective NCX-inhibitor SAR296968 inhibits atrial pro-arrhythmic activity and improves diastolic and contractile function in human atrial myocardium, which may have therapeutic implications, especially for treatment of HFpEF.

## 1. Introduction

The prevalence of heart failure (HF) is high and is expected to increase in the coming years due to demographic changes. Despite advances in heart failure therapy, morbidity and mortality in HF patients remain high, with an overall five-year mortality of 50%, which is even higher in end-stage HF [1,2]. HF is associated with systolic and diastolic contractile dysfunction caused by abnormalities of intracellular Ca^2+^ handling and structural remodeling [3]. Several targets related to the remodeling processes have been identified. For instance, sarcoplasmic reticulum (SR) Ca^2+^ -ATPase (SERCA) proteins are downregulated and associated with a decreased Ca^2+^ uptake capacity of the SR in the human heart [4,5].

In contrast, protein expression and activity of the cardiac sarcolemmal Na^+^/Ca^2+^ exchanger (NCX1) have been found to be increased in HF, making it even more effective in competing with decreased SERCA activity for cytosolic Ca^2+^ degradation [6,7]. In addition to HF, increased NCX expression is also observed in atrial fibrillation (AF) [8]. In both conditions, increased NCX expression has been associated with decreased contractility [9,10], and adenoviral overexpression of NCX has been shown to reduce contractility in rabbit myocytes [11]. The underlying mechanism involves impaired SR Ca^2+^ loading, resulting in decreased intracellular Ca^2+^ transients, increased diastolic Ca^2+^ levels on the cellular level, and decreased contractile force with diastolic dysfunction in vivo [7,12]. Importantly, up to 50% of HF cases are heart failure with preserved ejection fraction (HFpEF) [13].

Moreover, NCX seems to play a key role in impaired cardiac function in HFpEF [14,15,16]. Besides contractile function, NCX has also been implicated in developing ventricular and atrial arrhythmias [17,18]. NCX activity in forward mode can increase transient inward current resulting in delayed afterdepolarizations [19]. Recently, pharmacological NCX inhibitors improved cardiac function and displayed antiarrhythmic properties in guinea pigs and dog hearts [20,21]. Additionally, cardiac function in a murine HFpEF model was improved by NCX inhibition [14]. However, despite this clear evidence, no pharmacological NCX inhibitors with high specificity and potency have currently shown efficacy in humans.

The present study investigates the efficacy of a novel NCX inhibitor, SAR296968, developed for use in humans on intracellular Ca^2+^ homeostasis, pro-arrhythmic activity and contractile function in human atrial myocardium.

## 2. Materials and Methods

A detailed description of the methods utilized is provided in the Appendix A.

### 2.1. Human Myocyte Isolation

All procedures were performed according to the Declaration of Helsinki and were approved by the local ethics committee. All patients gave written informed consent prior to study inclusion. Biopsies of human right atrial appendages of patients undergoing aorto-coronary bypass grafting were used for chunk isolation of individual myocytes. Only elongated cells with cross-striations and without granulation were selected for experiments.

### 2.2. Preparation of SAR296968 Solutions and Storage Conditions

SAR296968 was solubilized in DMSO to a 3 mmol/L stock and kept at −20 °C. The addition of a working solution accomplished final dilution. For control experiments, the “vehicle” groups used equal amounts of DMSO. For all experiments, the maximum DMSO concentration during the experiments was 0.1%.

### 2.3. Measurement of Ca^2+^ Sparks 

Myocytes on laminin-coated recording chambers were loaded with 10 µmol/L fluo-4 acetoxymethylester in the presence of 0.02% (*w*/*v*) pluronic acid (30 min incubation), mounted on the stage of a laser-scanning confocal microscope (LSM5, Zeiss, Oberkochen, Germany) and superfused with normal Tyrode (NT) solution at room temperature. Fluo-4 was excited at 488 nm, and emission was collected through a 505 nm long-pass filter. Ca^2+^ sparks were analyzed using the SparkMaster plugin for ImageJ (Version 1.52) [22]. For some experiments, caffeine (10 mmol/L) was applied to induce rapid SR Ca^2+^ release, and the resulting transients were analyzed. 

### 2.4. Patch-Clamp Experiments

Myocytes were mounted on the stage of a microscope (Zeiss Axio Observer, Zeiss, Oberkochen, Germany). A ruptured-patch whole-cell current-clamp was used to measure cardiac membrane potential using microelectrodes (~5–7 MΩ). Access resistance was typically <15 MΩ after patch rupture. Action potentials were continuously elicited in current clamp configuration by square current pulses of 1–2 nA amplitude and 1–5 ms duration at variable basic cycle lengths. Signals were filtered with a 2.9 kHz Bessel filter and recorded with an EPC10 amplifier (HEKA Elektronik, Reutlingen, Germany). All experiments were conducted at room temperature.

### 2.5. Measurements of Contractility in Human Atrial Trabeculae 

Trabeculae were dissected from right-atrial appendage biopsies. For isometric force recordings, trabeculae were mounted in a chamber and connected to a force transducer. As previously described, trabeculae were superfused with Krebs–Henseleit solution and were electrically field-stimulated (1 Hz) [23]. Contractility was analyzed in steady-state. Post-rest behavior was assessed by measuring the force after a 30 s pause of electrical stimulation, whereas the ratio of the first contraction after the pause and the steady state before the pause was assessed. The developed force was normalized to the cross-sectional area of each trabecula (thickness × width × π/4) and expressed in mN/mm^2^.

### 2.6. Statistical Analysis

All data in graphs are expressed as mean patient data ± SEM. Continuous variables are expressed as mean ± SD or median (IQR) for baseline patient characteristics, and categorical variables as frequencies and percentages, respectively. Statistical analysis was conducted on mean patient data using SPSS version 28 (IBM, Armonk, NY, USA) and GraphPad Prism 9 (Graphpad Software, San Diego, CA, USA). Normality was assessed by Shapiro–Wilk normality test. For experiments comparing more than two groups of paired data, RM-mixed effects analysis with Bonferroni’s multiple comparisons test was performed. In selected experiments, tests for linear trends between groups were performed. For this, data was arranged with inhibitor concentrations in ascending order. All reported *p*-values are two-sided, and the threshold for significance was set at *p* < 0.05. 

## 3. Results

### 3.1. Patient Data

Table 1 depicts baseline patient characteristics for all investigated patients. Most of these patients were male. Many patients suffered from HFpEF, diagnosed according to current guidelines [24]. Besides coronary heart disease, arterial hypertension, diabetes, and atrial fibrillation were comorbidities.

### 3.2. SAR296968 Decreases Pro-Arrhythmic Activity in Human Atrial Cardiomyocytes 

The pharmacodynamic profile of SAR296968 has been described elsewhere [20]. To analyze the effect of SAR296968 on triggered atrial activity, confocal laser scanning microscopy was performed in Fluo-4 loaded myocytes. The mean data for Ca^2+^ spark frequency measured in human cardiomyocytes isolated from the right atrial appendage biopsies are displayed in Figure 1B. Compared to vehicle, the Ca^2+^ spark frequency (CaSpF) was significantly reduced in the presence of the NCX inhibitor SAR296968 from (in 100 µm^−1^ s^−1^) 2.679 ± 0.1949 (vehicle, mean ± SEM) to 1.952 ± 0.1859 (300 nM) and further to 1.802 ± 0.1491 (3 µM) (*p* = 0.048 vehicle vs. 300 nM and *p* = 0.0075 vehicle vs. 3 µM, the original confocal line scans depicted in Figure 1A). Furthermore, a linear trend between groups for a reduction in CaSpF was evident (*p* = 0.0015 for linear trend). Subgroup analysis revealed that the reduction of CaSpF was present in myocytes of HFpEF and non-HFpEF patients (Appendix A).

The SR Ca^2+^ leak and Ca content are related, and SR Ca^2+^ content can be measured through SR Ca^2+^ depletion by rapid application of caffeine. Interestingly, despite decreased CaSpF, SR Ca^2+^ content measured by caffeine transient amplitude (ΔF/F_0_) was significantly increased by SAR296968 at 3 µM from 8.926 ± 0.836 to 11.58 ± 1.302 (*p* = 0.007 vs. vehicle, Figure 1C, original transients shown in Figure 1D, averaged transients in the Appendix A). Interestingly, the subgroup analysis revealed that caffeine transient amplitude was significantly increased by SAR296968 in non-HFpEF patients (*p* = 0.038 for linear trend, Appendix A), while there was only a non-significant trend in HFpEF patients (*p* = 0.068 for linear trend). Of note, SAR296968 did not significantly alter Ca^2+^ spark width, amplitude, or total SR Ca^2+^ leak but significantly increased Ca^2+^ spark duration at 3 µM (Appendix A–D). 

### 3.3. SAR296968 Improves Contractility in Human Atrial Trabeculae

The increased SR Ca^2+^ content may improve contractile function because more Ca^2+^ is available for systolic Ca^2+^ release. To further analyze the effects of SAR296968, human atrial trabeculae were superfused with either vehicle or SAR296968, and tension was monitored as a function of time under regular electric field stimulation (1 Hz). Interestingly, in the presence of SAR296968, the diastolic tension was reduced from 2.432 ± 0.2194 mN/mm^2^ (vehicle) to 1.803 ± 0.2711 mN/mm^2^ (300 nM) and 1.670 ± 0.2552 mN/mm^2^ (3 µM) (Figure 2A, *p* = 0.0051 vehicle vs. 300 nM, *p* = 0.0039 vehicle vs. 3 µM, original recordings in Figure 2D). Moreover, there was a significant linear trend between groups (*p* = 0.0013 for linear trend). Intriguingly, this reduction in passive tension by SAR296968 was significant in HFpEF patients upon subgroup analysis (Appendix A, *p* < 0.001 vs. vehicle for all concentrations). 

Additionally, the developed tension as a measure of systolic contractile function was assessed. Although raw values did not significantly differ between groups (Appendix A) due to large variability between different trabeculae, the fractional difference of developed tension after exposure to SAR296968 (in % of baseline tension under the vehicle) was significantly increased at 3 µM (Figure 2B, *p* = 0.035 vs. vehicle). Additionally, there was a positive linear trend with increasing SAR296968 concentration (*p* = 0.018 for linear trend).

Congruent with the increase in SR Ca^2+^ content and reduction in spontaneous SR Ca^2+^ release (see above), the ratio of developed force after a 30 s pause of electrical stimulation to steady-state developed force, i.e., the “post 30 s rest ratio”, was significantly increased by exposure to SAR296968 from 0.79 ± 0.09895 (vehicle) to 1.192 ± 0.1682 (3 µM, Figure 2C, *p* = 0.04 vehicle vs. 3 µM). There was a positive linear trend with increasing SAR296968 concentration (*p* = 0.02 for linear trend). In contrast to diastolic tension and systolic developed force, SAR296968 did not significantly alter trabeculae relaxation kinetics, indicated by the time to reach 80% of baseline level (RT80) (Appendix A).

### 3.4. Electrophysiological Safety Parameters Remain Unaltered by SAR296968

Patch clamp experiments were performed on isolated human atrial cardiomyocytes to assess the potential safety of pharmacological NCX inhibition by SAR296968. The resting membrane potential was similar between groups (Figure 3A). Importantly, the action potential (AP) amplitude (Figure 3B) and AP duration at 90% (APD90) repolarization remained unchanged (Figure 3C, additional characteristics depicted in Appendix A). The original AP recordings are shown in Figure 3D. 

## 4. Discussion

To date, pharmacological NCX inhibition has been broadly tested in animal models, including rodents and large species [14,20,21]. However, to our knowledge, translation of specific NCX inhibitors to human cardiac tissue has not yet been performed. This study investigated the potential antiarrhythmic and contractile effects of a highly selective NCX inhibitor, SAR296968, in isolated human myocytes and multicellular experiments with atrial trabeculae.

### 4.1. Pharmacological Profile and Safety 

Previously, the safety of SAR296968 has been studied. In a recent study, SAR296968 did not cause relevant inhibition of I_Na_ or I_Ca_ in vitro nor prolongation of the QT interval in vivo while providing high pharmacological selectivity with an IC_50_ of 74 nM for human NCX1 [20]. In addition, Pelat et al. measured the NCX currents in guinea pig cardiomyocytes and Chinese Hamster Ovary cells expressing human NCX isoforms. NiCl_2_-sensitive currents (upon the voltage ramps from −120 mV to +60 mV) were analyzed and showed that SAR296968 inhibited both the NCX forward mode (at −90 mV E_m_) and the reverse mode (at +45 mV E_m_) with an IC_50_ for the forward mode of 34.9 nM (95% CI: 29.0–42.0) and 38.9 nM (95%CI: 30.6–49.5) for reverse mode in guinea pig cardiomyocytes [20]. At the highest dose studied by Pelat et al., the prolongation of PR interval in dogs was observed; however, this was less pronounced in HF subjects [20]. In the present study, the safety of pharmacological NCX inhibition with SAR296968 was confirmed by the measurement of resting membrane potential and action potential characteristics. The action potential upstroke velocity is a surrogate of peak sodium current and thus voltage-gated sodium current function [25]. Additionally, AP prolongation is related to arrhythmia development. Importantly, all AP characteristics remained unaltered by exposure to SAR296968. Although NCX is the main Ca^2+^ extrusion mechanism from the cell in cardiomyocytes and is therefore involved in repolarization [17], we did not observe APD prolongation in human atrial myocytes under NCX inhibition (Figure 3C).

### 4.2. SAR296968 Reduces Triggered Activity in Atrial Myocytes

An increased SR Ca^2+^ leak is a hallmark of heart failure (HF) [26], and is associated with the development of arrhythmias [23]. Ca^2+^ sparks occur during the diastole, and increased Ca^2+^ spark frequency and total Ca^2+^ leak, therefore, serve as arrhythmogenic triggers [27]. The NCX function results in net membrane depolarization due to the transport of one Ca^2+^ in exchange for 3Na^+^, which may facilitate the triggering of arrhythmias. Additionally, during conditions of increased diastolic SR leak, enhanced Ca^2+^ extrusion via NCX may further enhance this effect.

In HF, but also sleep-disordered-breathing and atrial fibrillation, enhanced Na^+^ influx contributes to increased intracellular Na^+^ concentrations [18,23,27]. In addition, there has recently been increasing evidence of increased intracellular Na^+^ concentration in HFpEF [14,15,16]. Increased Na^+^ disrupts NCX function and can facilitate Na^+^/Ca^2+^ exchange in reverse mode, which results in a Ca^2+^ influx into the cell [16,28,29]. The Ca^2+^ that enters the cell via reverse mode NCX could bind to the cytoplasmic surface of cardiac RyR2, increasing its open probability (P_o_) [30]. In fact, reverse mode NCX has been suggested to be crucially involved in the regulation of RyR P_o_ during cardiac excitation-contraction coupling [31,32]. Moreover, it has been shown that a substantial amount of cardiac NCX is expressed in a very close spatial relationship to cardiac RyR2 [33]. Especially in failing myocytes, reverse mode NCX may result in a substantial Ca^2+^ influx during the action potential [34]. Subsequently, the activation of Ca^+^/calmodulin-dependent protein kinase II (CAMKII) by reverse mode NCX has previously been described, also in human atrial cardiomyocytes and discussed in detail [26]. CAMKII then undergoes autophosphorylation, sustaining its activated state, and is able to phosphorylate cardiac RyR2, increasing its P_o_ and the frequency of Ca^2+^ sparks [35]. Thus, the inhibition of reverse-mode NCX by SAR296968 may have contributed to the reduced Ca^2+^ spark frequency observed in our experiments. 

A similar mechanism of indirect modulation of cardiac RyR P_o_ is also present after inhibition of cardiac voltage-gated Na channels [36]. Other drugs, including selective sodium channel blockers, have exhibited such indirect effects [27]. Finally, although unlikely due to the high pharmacological specificity of SAR296968, off-target effects which may have contributed to this reduction in Ca^2+^ spark frequency can never be completely ruled out.

Interestingly, in AF patients, spontaneous SR Ca^2+^ release events occur more frequently in atrial myocytes [37]. Moreover, Ca^2+^ spark frequency was also increased in myocytes in a murine HFpEF model [15]. Here, we investigated the effect of NCX inhibition on triggered activity in isolated human atrial myocytes. We observed a significant reduction in CaSpF through pharmacological NCX inhibition (Figure 1A,B). Therefore, reducing triggered activity via pharmacological NCX inhibition may be a novel antiarrhythmic strategy. Of note, despite reduced CaSpF, the calculated SR Ca^2+^ leak was not significantly reduced by SAR296968 (Appendix A). 

There are several explanations for this discrepancy. First, not all SR Ca^2+^ leak is detectable as Ca^2+^ -sparks leading to underestimation of total SR Ca^2+^ leak if Ca^2+^ leak was calculated using CaSpF [38,39]. Second, the increases in SR Ca^2+^ content (see below) can increase RyR2 open probability [38], a possible explanation for the increase in Ca^2+^ spark duration we observed at 3 µM (Appendix A). In fact, an increased Ca^2+^ spark duration has been predicted as a function of SR Ca^2+^ load in mathematical models [39]. Importantly, this prolonged Ca^2+^ spark duration may counteract the reducing effect on Ca^2+^ leak that comes from decreased CaSpF, as more Ca^2+^ is released per spark, which could explain an unchanged total Ca^2+^ leak.

### 4.3. SAR296968 Increases SR Ca^2+^ Content and Improves Contractility in Human Atrium

While the increased CaSpF is associated with increased triggered activity [27], it also results in SR Ca^2+^ loss and can reduce SR Ca^2+^ content [40]. Thus, we investigated the effect of pharmacological NCX inhibition on steady-state SR Ca^2+^ load. Interestingly, after exposure to SAR296968, caffeine transient amplitude was significantly increased at 3 µM, consistent with a higher SR Ca^2+^ content. Higher SR Ca^2+^ content corresponds to improved systolic contractility [32], and our results match previous results of investigations of NCX inhibition in animal models [14,21,41]. Accordingly, we were able to demonstrate that the fractional difference in contractile force measured in isolated atrial trabeculae at 1 Hz stimulation was significantly increased by NCX inhibition compared to vehicle (Figure 2B). 

Recently, in vivo experiments demonstrated that pharmacological NCX inhibition improved contractility in a canine HF model [20]. Interestingly, contractility was also improved for non-failing dogs [20]. Moreover, papillary and atrial muscle trabeculae from guinea pigs displayed increased contractile force upon NCX inhibition, and DAD-related arrhythmias were reduced [20]. In isolated cardiac muscle trabeculae, the ratio of the contractile force after a pause to the baseline force before pause (post-rest test) is an indicator of diastolic SR Ca^2+^ leak [23]. In HF, increased spontaneous SR Ca^2+^ release in the form of sparks due to enhanced sensitivity of RyR2 to cytosolic Ca^2+^ levels reduce SR Ca^2+^ content and thus impairs contractility [42]. Importantly, the post-30 s-rest ratio was significantly increased at 3 µM SAR296968 (Figure 2C). Therefore, reducing CaSpF and increasing SR Ca^2+^ content by NCX inhibition may reduce arrhythmia development and improve contractility in AF and HF patients. 

### 4.4. SAR296968 Improves Diastolic Function

Under the conditions of increased intracellular Na^+^ concentrations like heart failure, reverse mode NCX is favored, and the inhibition of reverse mode NCX may, thus, reduce diastolic Ca^2+^ levels and improve diastolic function [16]. Previously, pharmacological NCX inhibition with SEA0400 or ORM-11035 in a rat HFpEF model demonstrated improved diastolic function and remodeling in vivo [14,15]. Importantly, ORM-11305 did not alter relaxation kinetics at baseline, but relaxation in failing rat myocytes was improved [14]. Thus, in our study, SAR296968 significantly reduced diastolic tension at both 300 nM and 3 µM in human atrial trabeculae (Figure 2A), which implicates lower diastolic Ca^2+^ levels by reduced reverse mode NCX-dependent Ca^2+^ entry. 

On the other hand, both the sarcoplasmic reticulum Ca^2+^ ATPase (SERCA) and NCX compete for Ca^2+^ removal, and therefore SERCA function may be improved by NCX inhibition, especially under conditions of increased NCX activity such as HF and AF. The experimental evidence provided in our study underlines improved SERCA function, as SR Ca^2+^ content was increased by pharmacological NCX inhibition, as was the relative contractile force and post-rest contractile force, while simultaneously reducing pro-arrhythmic spontaneous Ca^2+^ release events. Moreover, at the doses investigated, SAR296968 did not negatively influence action potential characteristics. Prolongation of action potential duration could further increase diastolic Ca^2+^, but this is not the case with SAR296968. 

### 4.5. Translational Implications

The enhanced NCX expression has been reported in different cardiac diseases, including heart failure and atrial fibrillation [7,8]. NCX upregulation in these diseases was associated with reduced contractility in isolated trabeculae [9,10]. Therefore, pharmacological NCX inhibition was identified as a potential therapeutic target to improve contractility, but early inhibitors lacked specificity and potency [43]. Recently, NCX has been associated with impaired relaxation and reduced cardiac function in HFpEF and conducted animal studies were promising [14,15]. Additionally, NCX has been implicated in the generation of cardiac arrhythmias [17]. Moreover, in conditions of enhanced intracellular sodium concentration such as AF [18], reverse-mode NCX activity can increase spontaneous Ca^2+^ release events by activating Ca^2+^ -induced Ca^2+^ -release. In conclusion, NCX inhibition may be a potential novel therapeutic strategy to combat AF and improve contractility, especially in HFpEF patients. 

### 4.6. Limitations

Due to the limited availability of tissue, SAR296968 was the only NCX-inhibiting compound investigated in two different concentrations (300 nM and 3 µM) in this study. Although the detailed pharmacological profile regarding potency and selectivity of SAR296968 indicates a high selectivity for NCX [20], we cannot completely exclude off-target effects in our study. We did not analyze the effect of SAR296968 on human ventricular myocytes, which may limit the translation of these findings to ventricular pathologies. Nevertheless, our study suggests that SAR296968 application may also be useful to treat atrial remodeling, which is also present in many patients with HFpEF [44]. Finally, since Fluo-4-AM is a non-ratiometric dye, and we did not measure sarcolemmal caffeine-induced current, it was not possible to determine absolute cellular Ca^2+^ concentration.

## 5. Conclusions

In this study, we were able to show that SAR296968, a potent NCX inhibitor, significantly increased the developed force of human atrial trabeculae, while simultaneously significantly reducing diastolic tension. Moreover, the frequency of spontaneous SR Ca^2+^ release events were suppressed by pharmacological NCX inhibition. Additionally, the SR Ca^2+^ content was increased accordingly. Importantly, these effects were reached while maintaining a high inhibitor selectivity, and no effects on cardiac action potential characteristics were detected. These results demonstrate the potential of this novel compound developed for use in humans as a novel therapeutic pharmacotherapeutic strategy, especially for HFpEF patients.

## Figures and Tables

**Figure 1 biomedicines-10-01932-f001:**
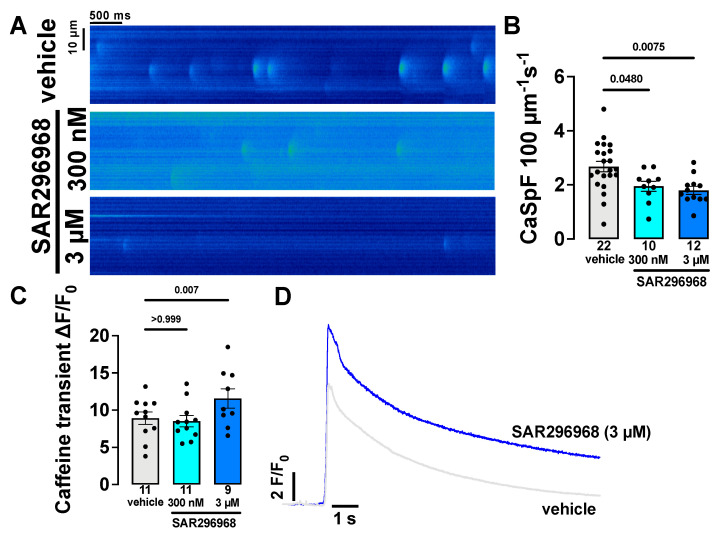
Ca^2+^ sparks measurements in isolated human atrial myocytes. The numbers indicated are numbers of patients. The data points represent individual patients. Each data point was calculated as a mean from several cells within each patient. Ca^2+^ spark frequency is reduced by SAR296968, original line scans are shown in (**A**), and mean data in (**B**). Fluo-4 transient amplitude after rapid caffeine application is shown in (**C**), the transient amplitude is significantly increased at 3 µM, original caffeine transients in (**D**). The *p*-values are denoted above the corresponding groups, RM-mixed effects analysis with Bonferroni post-hoc test.

**Figure 2 biomedicines-10-01932-f002:**
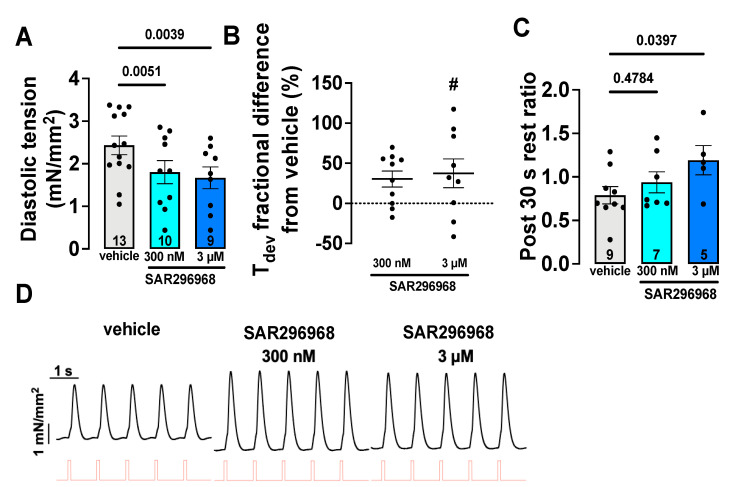
The numbers indicated are the numbers of patients. The data points represent individual patients. Each data point was calculated as the mean from several trabeculae within each patient. The mean data per patient shown for isolated human atrial trabeculae in steady-state (1 Hz) exposed to SAR296968. The diastolic tension is reduced by SAR296968 (**A**). The fractional difference in the developed tension compared to the vehicle was significantly increased (**B**), # indicates a significant difference from zero). Developed tension after 30 s rest (post-rest test) was significantly increased by SAR296968 at 3 µM (**C**). The original traces are shown in (**D**). The *p*-values are denoted above the corresponding groups, RM-mixed effects analysis with Bonferroni post-hoc test.

**Figure 3 biomedicines-10-01932-f003:**
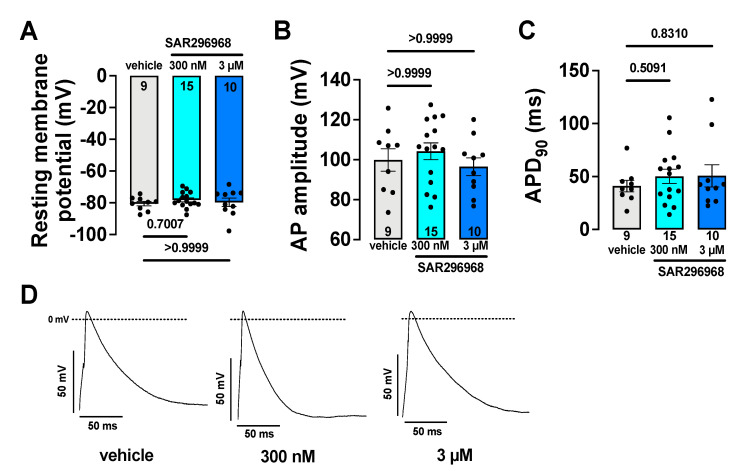
The numbers indicated are numbers of patients. The data points represent individual patients. Each data point was calculated as the mean from several cells within each patient. The mean data per patient for action potential characteristics were measured in isolated human atrial myocytes by the whole-cell patch clamp technique. (**A**) resting membrane potential, (**B**) AP amplitude, and (**C**) action potential duration (APD) at 90% of maximum repolarization were unaffected by SAR296968. The original traces are shown in (**D**). The *p*-values are denoted above the corresponding groups, RM-mixed effects analysis with Bonferroni post-hoc test.

**Table 1 biomedicines-10-01932-t001:** Baseline patient characteristics.

	n = 46
Male gender, n (%)	40 (87)
Age (years), mean ± SD	68.8 ± 7.7
Body mass index (kg/m^2^), median [IQR]	28.6 [6.6] n = 35
Coronary heart disease, n (%)	44 (95.7)
LVEF (%), median (IQR)	60 [14.5] n = 40
HFpEF, n (%)	24 (52.2)
Atrial fibrillation, n (%)	3 (6.5)
GFR (mL/min), mean ± SD	70.6 ± 21.5 n = 44
**Type of surgery**	
CABG, n (%)	44 (95.7)
Aortic valve replacement, n (%)	6 (13)
Mitral valve replacement, n (%)	1 (2.2)
**Cardiovascular risk factors**	
Arterial hypertension, n (%)	37 (88.1)
Hypercholesterolaemia, n (%)	19 (41.3)
Diabetes mellitus, n (%)	3 (7.7) n = 39

Values are presented as mean ± SD or median [IQR], respectively. Abbreviations: left-ventricular ejection fraction (LVEF), heart failure with preserved ejection fraction (HFpEF), glomerular filtration rate (GFR), coronary artery bypass graft surgery (CABG). HFpEF was diagnosed according to current ESC guidelines [24].

## Data Availability

The datasets included in the current study can be obtained from the corresponding author upon reasonable request, and after informed patient consent is obtained from study subjects.

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
