# Peer review of "SAR296968, a Novel Selective Na+/Ca2+ Exchanger Inhibitor, Improves Ca2+ Handling and Contractile Function in Human Atrial Cardiomyocytes"

_biomedicines, 2022, doi:10.3390/biomedicines10081932_

Round 1
Reviewer 1 Report
The paper "SAR296968, a novel selective Na+/Ca2+ exchanger inhibitor, improves Ca2+ handling and contractile function in human cardiomyocytes" (Hegner et al.) is quite suitable for publication. The methods and results in general very well described and the findings provide new insights to the pathophysiology of failing myocardium. I would recommend this paper for publication if the authors provide their comments on the following remarks and make requested modifications.
Page 1, Abstract, line 28: I suggest to use "control" instead of "vehicle". Acording to Methods, "vehicle" may indicate a saline without SAR296968 and with a certain amount of DMSO, but no this term was explicitly used in the Methods.
Page 3, line 120: Replace "In select experiments..." by "In selected experiments..."
Page 4, Table 1: Abbreaviations were not listed (while footnote says they were did before).
Page 4, lines 143-144: Please indicate measure units here for CaSpF as in Figure 1, i.e. 100/(microm*s).
Page 4, lines 142-145 and Page 5, Figure 1: Figure 1 contains the numbers in panels A, B, C. Seems these are the numbers of cells in "vehicle", "300 nM SAR296968", and "3 microM SAR296968"; however, the numbers are different. Does it mean that the effect of SAR296968 was tested in those cells which were not evaluated in "vehicle" condition? If so, why not to use the same cells first without loading by SAR296968 and then after incubation in SAR296968?
Page 4, lines 147-148: According to the selection of p-level significance, the difference between "vehicle" and "300 nM SAR296968" is not significant. The speculation about the presence of STRONG trend is not supported by the presented results.
Page 5, Figure 5: Please indicate in the figure legend the meaning of numbers shown in panels A, B, C. I suppose these are the numbers of cells used for analysis in each group.
Page 5, Figure 1, panel B: The result shown here indicates that the presence of high concentration of SAR296968 does not affect SR Ca leak while the effect was seen in a lower concentration. It does not conform well with the results shown in panel A, where Ca spark frequency was decreased in both concentrations. Can you propose an explanation?
Page5, Figure 5, panel C: Please indicate clearly whether the same cells were used to elicit caffeine-induced Ca-transient in "vehicle" and in presence of SAR296968. If no, the representative traces in Figure 1E may differ so greatly just due to the different initial Ca content in these two inpedendent cells. If different cells were used in all groups, you can demonstrate the effect of SAR296968 on caffeine-induced Ca-transient on whole Ca-transient by use of the averaged Ca-transients obtained for all cells in each group ("vehicle" and "3 microM SAR296968").
Page 5, Figure 5: Why not to present also the data for diastolic Ca levels of "vehicle" cells and the cells incubated in the presence of SAR296968? This may additionally prove the results shown in Figure 1A (Ca spark frequency) and Figure 1B (Ca leak from SR) and also can support the findings shown in Figure 2A (diastolic tension).
Page 5, lines 164-165: The sentence "This suggests that the decrease in CaSpF..." is not supported by the results as Ca leak from SR is not altered in 3 microM SAR296968. The sentence started with "A possible explanation for reduction in CaSpF..." was also not supported by corresponding measurements. These sentences must be deleted or removed to Discussion (in this case they should be carefully rephrased to conform with ACTUAL findings in this paper).
Page 5, line 171: I suggest to use "non-significant" instead of "minimal". However, it is better to say "did not significantly alter" about all three parameters shown in this Supplemental Figure 1.
Page 5, line 173, subheading: Please remove first appearance of "SAR-".
Page 5, lines 177-178 and Page 6, lines 183-184: The speculation about the positive lusitropism is not supported by any findings (passive tension is not related to lusitropy). On the contrary, Supplemental Figure 2B shows that the lusitropy was not altered in the presence of SAR296968.
Page 6, Figure 2, panel D: Abbreviation "CTRL" is used here while "Veh" is used in the previous figure and in Figure 2A-C. Please consider the suggestion to use "control" everywhere in the text (instead of "vehicle").
Page 6, Figure 2, panel B: How was it possible to compare statistically the fractional increase in active tension in the presence of SAR296968 vs. "vehicle" (as indicated by symbol "#"), if to produce the fractional values you was needed to divide the peak tension to corresponding peak tension in "vehicle" condition? Please explain the approach used to evaluate the significance.
Page 6, line 207, subheading: Please remove first appearance of "SAR-".
Page 7, lines 244-245: "...NCX is the main Ca2+ extrusion mechanism..." - I suggest to add "from cell" to avoid mixing with "Ca extrusion from cytosol" (SERCA is the main contributor in the latter case).
Page 8, line 264: I suggest to change "...it has been suggested..." to "...it has been shown...".
Page 8, lines 288-289: Again, I suggest to avoid this speculation. Figure 1B and corresponding p-value show that there was no trend of the effect of SAR296968.
Page 9, lines 310-312: Please update "...the ratio of the contractile force after a pause..." by "...the ratio of the contractile force after a pause to the baseline force before pause..."
Page 9, line 315: Again, I cannot agree that the speculation "This evidence corresponds to decreased SR Ca2+ leak.", which is linked to the previous sentence, can be used in regard to the data shown in Figure 1B. The data on Figure 1B does not show that SAR296968 affected SR Ca leak.
Reference list: It looks to be a little bit longer for a research article, I would suggest to shorten it to 40-45 items.
Author Response
Please see the attachment "Response to the Editor and Reviewers", section "Reviewer 1", thank you.

Reviewer 2 Report
Comments to the Authors
The manuscript entitled “SAR296968, a novel selective Na+/Ca2+ exchanger inhibitor, improves Ca2+ handling and contractile function in human cardiomyocytes“ (Manuscript ID: biomedicines-1824065) describes the beneficial actions of the novel NCX inhibitor, SAR296968 on human right atrial muscle and isolated cardiomyocytes. It shows the effects of SAR296968 on calcium homeostasis by measuring Ca sparks (increase of their frequency) and estimates SR calcium content by caffeine induced calcium signals (elevation of calcium content of the stores in the higher dose). It demonstrates that increase of isometric force and the reduction of diastolic tension by SAR296968 in human right atrial trabeculae. Moreover, the atrial action potential is not influenced by the NCX blocker which supports the lack of its effect on major cardiac ion currents. The study is interesting, well-designed with high quality manuscript. The manuscript is almost free of misspellings and easy to follow. The graphs are fairly good quality although their resolution could be improved. The number of experiments is sufficient and the analysis of results is satisfactory. The conclusions are mostly supported by the results, which are compared with the literature. There are 58 references in a nicely organized manner.
Major comments/questions/suggestions:
1. My biggest concern is that the study suggests that SAR296968 application can be useful in patients with HFpEF too and not only in AF, despite testing the compound only in atrial but not in ventricular preparation. In the cited references also mainly ventricular preparations were used. I think this should be emphasized in the manuscript (in the discussion (e.g limitation of the study)) and in its title too.
2. Also, the study would be more valuable if one can be sure that the detected effects are indeed due to the NCX blockade developed by SAR296968 application. Did you try to reverse the action of the drug using washout or by comparing with the application of another NCX blocker?
3. Several times dose-dependent effects are mentioned based on only two applied SAR296968 concentrations. How reliable is stating that based on only so small number of doses? See Figure 1C where the smaller dose was identical (or smaller) compared to control and increase was only seen with the large dose.
4. Have you determined the spatial characteristics of calcium sparks or just their frequency? Was the location of spark development influenced by SAR296968 treatment or just their overall frequency?
5. What is the unit of SR calcium content (line 162)? Did you try determining the SR calcium content by measuring the integral of the ion current accompanying the transient?
6. Authors claimed that “in the presence of SAR296968, a positive lusitropic effect was observed.” (line 178) based on the reduction of diastolic tension. Positive lusitropic effect is mentioned also in line 183 but positive lusitropic effect is usually referred to the rate of relaxation and not the value of diastolic tension so these should be corrected especially because of the lack of SAR296968 action on relaxation kinetics (see RT80 on supplementary fig 2B).
7. The values of APD50 and APD90 are very small, even atrial APs should be longer (see for example DOI: 10.1016/j.hrthm.2015.10.003). What can be the reason for these small APD values? Can these values suggest a poor quality of atrial myocytes?
Minor comments/questions/suggestions:
1. What is the unit of CaSpF? It is missing from the text and it says µm-1s-1, which I do not understand as frequency is usually the number of observed sparks per unit time. Please clarify that.
2. The meaning of the dotted line is not given on figure 3D, please add that.
3. On the same panel, the original AP trace in 3 µM SAR296968 is not a fortunate choice as the peak and AP amplitude seems to be bigger despite the similar AP amplitude values of figure 3B.
4. The number of experiments with 3 µM SAR296968 on figure 1B is smaller compared to that of figure 1A (11 vs 12). The same applies for control on supplementary figure 1A and 1B (22 vs 21) and supplementary figure 2A and 2B (12 vs 13). What is the reason of this?
5. The number of experiments is not indicated on any panels of supplementary figure 4 making that different compared to all other figures so I suggest to indicate those.
6. The text part of the Supplemental Material does not contain too much extra information compared to the Material and Methods chapter in the manuscript, there is a large amount of repeated description.
Misspellings:
1. In line 120: “In select experiments…” should be corrected to “In selected experiments...”.
2. In line 144: “300nM” should be corrected to “300 nM” and please check and correct those (300nM and 3µM) in the entire manuscript including on the figures.
3. In line 155: “Transient” should be corrected to “transient”.
4. In line 207: “by SAR-SAR296968“ should be corrected.
Based on my previous comments I recommend at least a thorough minor revision of the manuscript.
Author Response
Please see the attachment "Response to the Editor and Reviewers", section "Reviewer 2", thank you.
